# Ecthyma Gangrenosum Secondary to Methicillin-Sensitive *Staphylococcus aureus* in an Atopic Child with Transient Neutropenia: A Case Report and Review of the Literature

**DOI:** 10.3390/diagnostics12071683

**Published:** 2022-07-11

**Authors:** Ko-Chun Fang, Fang-Ju Lin, Chih-Ho Chen, Yi-Ning Huang, Jui Lan, Han-Chi Tseng, Yi-Chuan Huang

**Affiliations:** 1Department of Education, Kaohsiung Chang-Gung Memorial Hospital and Chang Gung University College of Medicine, Kaohsiung 83301, Taiwan; erickfang1996@gmail.com; 2Department of Pediatrics, Kaohsiung Chang-Gung Memorial Hospital and Chang Gung University College of Medicine, Kaohsiung 83301, Taiwan; candy731731@cgmh.org.tw (F.-J.L.); cheer0613@cgmh.org.tw (C.-H.C.); b0102029@cgmh.org.tw (Y.-N.H.); 3Department of Pathology, Kaohsiung Chang-Gung Memorial Hospital and Chang Gung University College of Medicine, Kaohsiung 83301, Taiwan; blueray@cgmh.org.tw; 4Department of Dermatology, Kaohsiung Chang-Gung Memorial Hospital and Chang Gung University College of Medicine, Kaohsiung 83301, Taiwan; perkyjoy@cgmh.org.tw

**Keywords:** ecthyma gangrenosum, *Staphylococcus aureus*, transient neutropenia, atopic dermatitis

## Abstract

In addition to *Pseudomonas aeruginosa*, other organisms including *Staphylococcus aureus* have been reported to have associations with ecthyma gangrenosum (EG). There are very limited reports of *Staphylococcus aureus* EG causing systemic symptoms in an immunocompetent child. We present the case of an atopic child with transient neutropenia developing characteristic skin lesions of EG. Culture of the skin wounds yielded methicillin-susceptible *Staphylococcus aureus* (MSSA), and incisional biopsy of the skin lesions revealed aggregates of Gram-positive cocci at the subepidermal area and necrotic vasculitis but without perivascular bacterial invasion. In the literature review, seven cases of *Staphylococcus aureus* EG were reported, and only two were pediatric cases. From this case, we emphasize the importance of early culturing for microorganisms in cases presenting with EG. When toxin-mediated systemic symptoms accompany EG-like skin lesions, MSSA should be considered in an atopic child with transient neutropenia.

## 1. Introduction

Ecthyma gangrenosum (EG) is characterized by hemorrhagic papules with a violaceous rim developing into bullae, ulcers, and necrotic plaques [1]. It was initially described in association with *Pseudomonas aeruginosa* infection. Other organisms including *Staphylococcus aureus* have been reported to have associations with EG [2,3]. EG mostly occurs in immunocompromised patients. There are limited reports of *Staphylococcus aureus* EG causing systemic symptoms in children. We were presented with a one-year-old boy with a history of atopic dermatitis and neutropenia who developed characteristic skin lesions of EG secondary to methicillin-susceptible *Staphylococcus aureus* (MSSA) infection.

## 2. Case Presentation

A one-year-old boy with a history of atopic dermatitis was presented to our emergency department with a two-day history of fever and erythematous papules over the excoriated skin of the left arm, chest, abdomen, and right thigh (Figure 1a). There was no sign or symptom of respiratory tract infection. Laboratory studies were notable for a moderate neutropenia (absolute neutrophil count of 855 cells per μL, reference value, >1500) and an elevated C-reactive protein (CRP) level of 100.7 mg/L (reference value, <5). As cellulitis was initially diagnosed, he was admitted to the pediatric ordinary ward and intravenous oxacillin was initiated empirically. He was also found to have tarry stool and muddy diarrhea. On hospital day 2, the erythematous papules progressed to necrotic eschar with surrounding erythema (Figure 1b). Intravenous oxacillin was escalated to intravenous ceftazidime for coverage of *Pseudomonas aeruginosa*. However, he became drowsy and the skin lesions progressed on hospital day 4 (Figure 1c). Necrotizing fasciitis was suspected and he was transferred to the pediatric intensive care unit (PICU).

At the PICU, ceftazidime was shifted to cefepime, amikacin, and teicoplanin. Cultures of skin lesions yielded MSSA. Incisional biopsy of necrotic bullae revealed aggregates of Gram-positive cocci (GPC) at the subepidermal area (Figure 2a) and necrotic vasculitis without perivascular bacterial invasion (Figure 2b). Cultures of blood, urine, and cerebrospinal fluid were all sterile. Culture of stool did not yield *Pseudomonas aeruginosa*. His consciousness and activity improved gradually, and his skin lesions began to crust, desquamate, and heal. Antibiotics were then de-escalated to monotherapy with cefepime. On hospital day 11, laboratory studies revealed a white blood cell count of 6100/mm^3^, an absolute neutrophil count of 488/mm^3^, and a C-reactive protein level of 16 mg/L. The flow charts of presenting symptoms, laboratory data, and treatment courses in the hospital days are presented in Figure 3. An immunodeficiency study of T cell, B cell, lymphocyte proliferation, chemotaxis, and macrophage was unremarkable. A whole exome gene survey, especially the ELANE gene, for the detection of neutropenia showed a negative result. After completing 2 weeks of cefepime, he was discharged home with oral cephalexin.

## 3. Discussion

In our one-year-old male case with atopic dermatitis and transient neutropenia, his skin rash progressed rapidly from maculopapules to central necrosis then eschar formation. The clinical differential diagnosis in our case included EG associated with invasive pathogens: *Pseudomonas*, *Staphylococcus* sp., fungus, herpes simplex virus, and disseminated varicella zoster virus. Cultures of skin lesions yielded MSSA in our case.

Based on a literature review in PubMed between 1990 and 2021, using keywords of ecthyma gangrenosum and *Staphylococcus aureus*, we observed that there are only seven reported cases of EG associated with *Staphylococcus aureus* infection (Table 1) [1,4,5,6,7,8,9]. Among them are four cases with EG secondary to MRSA and three to MSSA infection; however, there are only two pediatric cases. Of the two pediatric cases [5,8], one was an immunocompromised eight-month-old infant initially presenting with febrile seizure and skin rash, which developed into EG secondary to MRSA infection. The other was a 15-month-old healthy child presenting with fever and flea-bite skin lesions, which progressed to EG due to MSSA infection. To the best of our knowledge, our patient is a rare case of EG secondary to MSSA in an atopic child with transient neutropenia.

*Pseudomonas aeruginosa* is the most common etiology documented in EG [6]. In children with EG, physicians must be aware that an underlying immunodeficiency might be present [10]. Predisposing factors include neutropenia, malignancy, burns, malnutrition, and tuberculosis [8]. It is important to identify any underlying hematological abnormalities, including variants of neutropenia such as chronic, cyclic, and transient neutropenia, the last of which is most reported in association with EG [11].

Transient neutropenia was finally diagnosed in our patient. As we know, many of the underlying causes of transient neutropenia are reversible. The causes can be associated with infectious and noninfectious diseases. Among the infectious diseases, bacterial infections are associated with neutropenia. On the other hand, the non-infectious causes of transient neutropenia are further divided into inflammatory and autoimmune disease [12]. In our case, we speculate that transient neutropenia was the result of bacterial infections which led to failure of production of neutrophils in the bone marrow or from their peripheral destruction [13,14]. Nevertheless, congenital or cyclic neutropenia should also be ruled out. Furthermore, congenital or cyclic neutropenia, which has been only rarely reported within the frame of EG [15], is now considered an autosomal dominant disease caused by ELANE gene mutations [16]. In our case, the whole exome survey for congenital or cyclic neutropenia, especially for the detection of the ELANE gene, showed a negative result.

EG is usually diagnosed clinically, but the causal pathogens should be confirmed by skin and, if indicated, blood cultures [17]. For the seven reported cases, the skin lesions had necrotic-appearing centers with surrounding erythema and soft tissue induration. In contrast, our case was the first one with underlying atopic dermatitis, initially presenting itchy skin rash. During flares of atopic dermatitis, *Staphylococcus aureus* is frequently isolated from the skin and many species that produce inhibitors of *Staphylococcus aureus* growth decline [18]. Under the scenario, we speculate that the increased *Staphylococcus aureus* on the skin of patients with atopic dermatitis might increase the risk of developing *Staphylococcus aureus* EG.

In our case, the organism was grown from culture of the skin lesion but not in blood cultures. Of the seven reported cases, there were only two cases with positive blood culture, but GPC were pathologically noted within the dermis in most cases. In our case, we clearly demonstrated aggregates of GPC at the subepidermal area of the skin lesion and necrotic vasculitis but without perivascular bacterial invasion into the vessels.

According to skin and soft tissue infection guidelines [19,20], intravenous antibiotic therapy in febrile neutropenic patients should be given for 7 to 14 days. Four of the seven reported cases were treated with antibiotics by using vancomycin alone or in combination with others. With the impression of EG possibly caused by *Pseudomonas aeruginosa* infection, we administered ceftazidime; however, the disease progressed. Therefore, the antibiotic was shifted to the combination therapy of intravenous cefepime, amikacin, and teicoplanin. Afterward, the pathogen of MSSA was pathologically identified and we de-escalated the antibiotics to cefepime alone. It is a clinical dilemma because young infants and toddlers are relatively immunodeficient and vulnerable to *Pseudomonas aeruginosa* sepsis complicated with EG.

## 4. Conclusions

From this case, we emphasize the importance of early culturing for microorganisms for cases presenting with EG. When toxin-mediated systemic symptoms accompany EG-like skin lesions, MSSA should be considered in an atopic child with transient neutropenia.

## Figures and Tables

**Figure 1 diagnostics-12-01683-f001:**
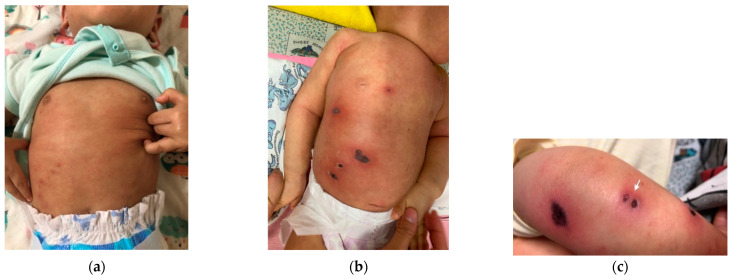
The initial presentation of the skin lesions revealed erythematous papules over the excoriated skin of chest and abdomen (**a**); the erythematous papules progressed to necrotic eschar with surrounding erythema (**b**); more progressive papulonodular lesions with necrosis (white arrow) were observed on limb skin (**c**).

**Figure 2 diagnostics-12-01683-f002:**
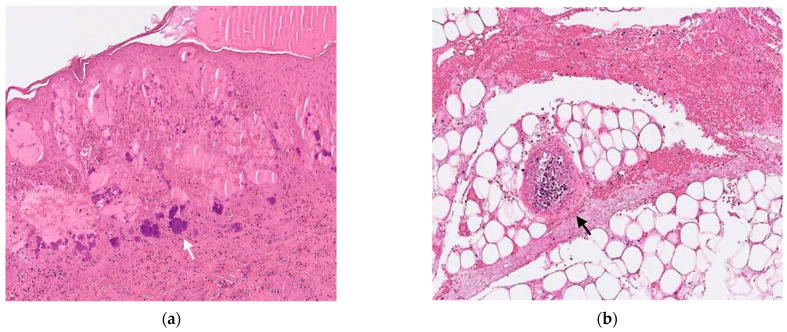
The pathologic images (Gram stain, ×200) of the incisional biopsy of the papulonodular skin lesion reveal (**a**) aggregates of Gram-positive cocci (white arrow) at subepidermal area and (**b**) necrotic vasculitis (black arrow) without perivascular bacterial invasion (Hematoxylin and Eosin stain, ×200).

**Figure 3 diagnostics-12-01683-f003:**
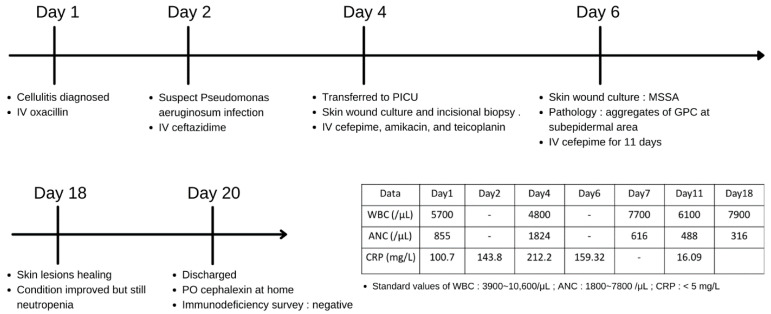
The flow charts of presenting symptoms, laboratory data, and treatment courses in the hospital days.

**Table 1 diagnostics-12-01683-t001:** Case reports of patients with Staphylococcus aureus ecthyma gangrenosum.

Author (Date)	Underlying Medical Condition	Initial Presentation	Blood Cultures	Lesion Cultures	Bacteria in Pathology	Antibiotics	Outcome
Pechter et al. (2012)	8-month-old infant with transient neutropenia	Febrile seizure, acute otitis media and skin rash	Negative	Culture: MRSA	GPC at epidermal area	IV vancomycin + cefepime + amikacin- > vancomycin	Discharged
Song et al. (2015)	Healthy 15-month-old girl	Fever and “flea bite-like lesions” on her chest	Negative	Culture: MSSA	Not observed	IV doxycycline- > PO cephalexin	Discharged
Chang et al. (2012)	35-year-old with leukemia	Tender and hemorrhagic skin lesions	Negative	Culture: MRSA	GPC in the dermis	IV vancomycin	Discharged
Ungprasert et al. (2013)	40-year-old male with AIDS	Painful lump in the right side of his neck	Negative	Prior skin abscess: MRSA	No biopsy	PO linezolid	Discharged
Ivanaviciene et al. (2015)	54-year-old female with SLE and metastatic gastric cancer	Fever and painful skin lesions	Negative	Culture: MSSA	GPC within intraepidermal area	IV vancomycin + nafcillin	Discharged
Shah et al. (2021)	A 62-year-old male with hypertension and hyperlipidemia	Scattered skin rash	Positive	Culture: MSSA	GPC in dermis	IV vancomycin + cefepime then- > nafcillin	Died of multiple organ failure
Sen et al. (2009)	69-year-old male with COPD	Mottled skin lesions	Positive	Culture: MRSA	No biopsy	IV ampicillin/sulbactam + meropenem + teicoplanin	Died of septic shock
Current presenting case	1-year-old boy with atopic dermatitis and transient neutropenia	Fever and itchy skin wounds	Negative	Culture: MSSA	GPC at subepidermal area	IV cefepime + amikacin + teicoplanin- > cefepime	Discharged

COPD, chronic obstructive pulmonary disease; MRSA, methicillin-resistant Staphylococcus aureus; AIDS, acquired immunodeficiency syndrome; SLE, systemic lupus erythematosus; MSSA, methicillin-sensitive Staphylococcus aureus; GPC, Gram-positive cocci; IV, intravenous; PO, oral administration of medication.

## Data Availability

The data are not publicly available due to ethical restrictions.

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
