# Peer review of "Ecthyma Gangrenosum Secondary to Methicillin-Sensitive Staphylococcus aureus in an Atopic Child with Transient Neutropenia: A Case Report and Review of the Literature"

_diagnostics, 2022, doi:10.3390/diagnostics12071683_

Round 1

Reviewer 1 Report

The current study reported an unusual case of ecthyma gangrenosum associated with Methicillin-Sensitive Staphylococcus aureus (MSSA) in an immunocompetent child with transient neutropenia. The patient history, diagnosis, etiology investigation and treatment courses were all very documented. I found the paper to be overall well written and felt confident that the authors performed careful observation and thorough investigation.

The discussion section is very well written and interesting. Ecthyma gangrenosum is commonly associated with Pseudomonas aeruginosa infection. The author observed and reported a pediatric case associated with Staphylococcus aureus. They further discussed the etiology of ecthyma gangrenosum based on the previously published literature. They emphasized the importance of early culturing for microorganisms presenting in EG to identify the pathogen. 

Author Response

Reviewer 1

Comments and Suggestions for Authors

The current study reported an unusual case of ecthyma gangrenosum associated with Methicillin-Sensitive Staphylococcus aureus (MSSA) in an immunocompetent child with transient neutropenia. The patient history, diagnosis, etiology investigation and treatment courses were all very documented. I found the paper to be overall well written and felt confident that the authors performed careful observation and thorough investigation. The discussion section is very well written and interesting. Ecthyma gangrenosum is commonly associated with Pseudomonas aeruginosa infection. The author observed and reported a pediatric case associated with Staphylococcus aureus. They further discussed the etiology of ecthyma gangrenosum based on the previously published literature. They emphasized the importance of early culturing for microorganisms presenting in EG to identify the pathogen. 

Answer: Thanks for kindly comment.

Reviewer 2 Report

This is a very interesting case report heightening awareness of the severity of an infection even without immunosuppression.

Minor comments:

1. Would you consider commenting on why this severe infection with transient neutropenia could arise in an otherwise healthy infant. Any other predisposing factors? In another case you mention insect bites. Did your case begin with something similar? Or was the atopic eczema poorly controlled? Infections or impetigo in caregivers?

2. When you give blood work values, would you consider giving norws for the laboratory you use? Or at least mention which are not normal?

Author Response

Reviewer 2

Comments and Suggestions for Authors. This is a very interesting case report heightening awareness of the severity of an infection even without immunosuppression.

Minor comments: 

  1. Would you consider commenting on why this severe infection with transient neutropenia could arise in an otherwise healthy infant. Any other predisposing factors? In another case you mention insect bites. Did your case begin with something similar? Or was the atopic eczema poorly controlled? Infections or impetigo in caregivers?

Answer: Thanks for the insightful comment. As mentioned in the Discussion, we have stated that the possible predisposing factor of the disease in our patient is associated with the poorly controlled atopic dermatitis. We stated that “During flares of atopic dermatitis, Staphylococcus aureus is frequently isolated from the skin and many species that produce inhibitors of Staphylococcus aureus growth decline. Under the scenario, we speculate that the increased Staphylococcus aureus on the skin of patients with atopic dermatitis might increase the risk of developing Staphylococcus aureus EG.” We did not find any relationship of the disease with insect bites or infection from caregivers in the case.

  1. When you give blood work values, would you consider giving norms for the laboratory you use? Or at least mention which are not normal?

Answer: Thanks for the kindly suggestion. We added the standard values of the laboratory data in the Figure 3.